# CELEB trial: Comparative Effectiveness of Lung volume reduction surgery for Emphysema and Bronchoscopic lung volume reduction with valve placement: a protocol for a randomised controlled trial

Sara Buttery,[1] Samuel V Kemp,[1] Pallav L Shah,[1] David Waller,[2] Simon Jordan,[1] John T Lee,[3] Winston Banya,[1] Michael C Steiner,[4] Nicholas S Hopkinson[1]

[1]NIHR Respiratory Disease, Biomedical Research Unit, Royal Brompton and Harefield NHS Foundation Trust, Imperial College London, London, UK
[2]Thorax Centre, Barts Health NHS Trust, London, UK
[3]School of Public Health, National University of Singapore, Singapore
[4]Leicester Respiratory Biomedical Research Unit, University Hospitals of Leicester NHS Foundation Trust, Leicester, UK

**Correspondence to**
Dr Nicholas S Hopkinson;
n.hopkinson@ic.ac.uk

## ABSTRACT

**Introduction** Although lung volume reduction surgery and bronchoscopic lung volume reduction with endobronchial valves have both been shown to improve lung function, exercise capacity and quality of life in appropriately selected patients with emphysema, there are no direct comparison data between the two procedures to inform clinical decision-making.

**Methods and analysis** We describe the protocol of the CELEB study, a randomised controlled trial which will compare outcomes at 1 year between the two procedures, using a composite disease severity measure, the iBODE score, which includes body mass index, airflow obstruction, dyspnoeaand exercise capacity (incremental shuttle walk test).

**Ethics and dissemination** Ethical approval to conduct the study has been obtained from the Fulham Research Ethics Committee, London (16/LO/0286). The outcome of this trial will provide information to guide treatment choices in this population and will be presented at national and international meetings and published in peer-reviewed journals. We will also disseminate the main results to all participants in a letter.

**Trial registration number** ISRCTN19684749; Pre-results.

### Strengths and limitations of this study

► This is a randomised controlled trial.
► There is a strict patient selection criterion for this trial based on previous research.
► Currently there is equipoise between the two interventions being compared.
► Due to the nature of the interventions it is not possible to double-blind the trial.
► It may be difficult to obtain longer term outcomes as patients may cross over after the 12-month period of the trial.

## INTRODUCTION

Chronic obstructive pulmonary disease (COPD) is a common and disabling condition which is now the third largest cause of death worldwide.[1] Breathlessness, exercise limitation and mortality in COPD are all associated with increased lung volumes occurring due to airflow obstruction and increased lung complicance.[2] The condition is progressive and despite optimum care including smoking cessation, pharmacotherapy and pulmonary rehabilitation, many patients remain breathless and limited in everyday activities.[3] [4] Surgical and bronchoscopic approaches to lung volume reduction are available which can bring substantial benefits in appropriately selected individuals, though both are also associated with some risk.

Lung volume reduction surgery (LVRS) involves removing the worst affected area of emphysematous lung, which means that the remaining healthier and less compliant lung can function more effectively, with the respiratory muscles working at less of a mechanical disadvantage.[5] LVRS has been shown to improve survival, exercise capacity and quality of life in appropriately selected patients with heterogeneous emphysema and poor exercise capacity,[5–9] and is therefore recommended in national and international guidelines for the management of COPD.[1] [10] [11] However, uptake has been limited, due in part to exaggerated concerns about surgical morbidity and mortality.[12] In modern clinical practice, morbidity and mortality from the procedure are substantially lower[9] [13] than was the case in trials conducted around the turn of the century.[5] [6]

**BMJ**

An alternative approach to resecting the target area of lung is to place endobronchial valves occluding the airways supplying a particular lobe. This form of bronchoscopic lung volume reduction (BLVR) can produce lobar atelectasis, and is intended to achieve similar benefits to LVRS but with less morbidity.[14–24] The approach is only effective in the absence of interlobar collateral ventilation (CV).[23 25] If this is present, air can enter the target lobe from an adjacent lobe and atelectasis does not occur. In patients with a heterogeneous pattern of emphysema and intact interlobar fissures, valve placement can produce significant improvements in lung function, exercise capacity and health status.[23 26–28] Yet complications do occur, in particular pneumothorax which occurs in up to 30% of cases[28] and can on occasion be fatal, as well as exacerbation-like events. Valve expectoration or misplacement can necessitate additional procedures.

Patients with heterogeneous emphysema and intact interlobar fissures may benefit from either BLVR or LVRS, but there are no direct comparison data on the relative value of the two procedures to guide clinical decision-making.

The CELEB study is a multicentre, randomised controlled parallel group trial that opened for recruitment in October 2016. Patients who are considered to be suitable candidates for both forms of targeted lung reduction therapy will be randomised to either BLVR or LVRS. The primary aim of the CELEB study is to determine whether LVRS procedures can achieve a greater health benefit at 1 year than BLVR. This will be determined using change in the iBODE index, a composite measure of COPD severity (made up of body mass index (BMI), airflow obstruction, dyspnoea and exercise capacity (the incremental shuttle walk test)).[29] Change in BODE score has been found to be a good predictor of survival in LVRS patients.[30] Health resource utilisation will also be compared between treatment arms. Clinical efficacy will be assessed over a 12-month period. The study is set up as a superiority trial and will be conducted, analysed and reported according to the Consolidated Standards of Reporting Trials statement for randomised controlled trials. This paper will outline the protocol for the CELEB study (version 2.0, 19 February 2016).

## METHODS AND ANALYSIS
### Participants
Patients will be recruited from outpatient clinics at UK hospital sites which have an established multidisciplinary team (MDT) meeting dedicated to identifying suitable candidates for LVR.[31] Patients will be eligible for the study if they fit the following criteria: significant airflow obstruction (forced expiratory volume in 1 s ($FEV_1$) <60% predicted), limiting breathlessness (Medical Research Council (MRC) dyspnoea score >3), significant hyperinflation (total lung capacity >100% predicted, residual volume (RV) >170% predicted) and are considered, using CT and perfusion scan, to have a heterogeneous emphysema pattern with intact interlobar fissures (>90%). They must also not have smoked for at least 3 months. Patients fitting these criteria will be identified from outpatient clinics and put forward for discussion at the MDT meeting. Patients will be excluded if they have pulmonary fibrosis or any other major comorbidity that could affect survival or mean that LVR procedures are unlikely to be effective. Patients with $PaO_2$ <7.0 kPa and/or $PaCO_2$ >7.0 kPa will also be excluded. The MDT will make a decision on whether a patient is suitable for both interventions and if so, that there is equipoise between the treatment options. It is only after this is agreed that the screening visit will be arranged and consent taken.

### Interventions
LVRS will be carried out by a thoracic surgeon under general anaesthetic using a unilateral video assisted thoracoscopic surgery (VATS) approach. The precise operation will be at the discretion of the surgeon, intended to remove the most emphysematous area of lung, leaving the best quality lung behind and minimising air leaks and other complications. The patients will initially go to the high dependency unit postoperatively, and postoperative management will include attention to wound discomfort, management of chest drains and prompt mobilisation. Patient management, discharge and initial follow-up will be determined by their clinical team.

BLVR will be performed via bronchoscopy by an operator experienced in placing endobronchial valves. The procedure will be performed under conscious sedation or general anaesthetic if necessary, with endobronchial valves placed to occlude the target lobe. A chest X-ray will be performed 1 hour after procedure and participants will spend three nights after procedure as an inpatient routinely in case a pneumothorax occurs. They will be discharged with written advice about the signs of pneumothorax and what to do if these occur. Procedures including valve adjustment and replacement will be permitted to ensure treatment is optimised.

Participants will be followed up in clinic routinely about 3 months after discharge. Other clinic attendances will depend on clinical need with further investigations and procedures determined by their responsible clinician. Patients will receive a phone call from the study team at 1 month and at 6 and 9 months. There will be a final clinic visit at 12 months. A summary of the trial assessment is shown in table 1.

### Outcomes
The primary outcome will be the change in iBODE score[32] at 12 months from baseline. The iBODE incorporates the incremental shuttle walk test (iSWT), BMI, $FEV_1$ and the MRC dyspnoea score. Possible scores range from 0 to 10, with 10 with increasing scores associated with worsening mortality. The primary endpoint measures will be performed by staff blinded to treatment allocation and patients will be asked not to reveal this in order to reduce bias.

**Table 1** Summary of trial assessments

| | Visit 1 Baseline | Visit 2 Chartis measurement | Visit 3 Admission for procedure | Visit 4 1 month safety phone call | Visit 5 3 months' follow-up | 6 months | 9 months | Visit 6 1 year follow-up |
|---|---|---|---|---|---|---|---|---|
| History, physical exam | x | | | | x | | | x |
| Informed consent | x | | | | | | | |
| FFM | x | | | | | | | x |
| MRC score | x | | | | x | | | x |
| CAT score | x | | | | x | | | x |
| Full pulmonary function* | x | | | | x | | | x |
| CT thorax | x | | | | x | | | x |
| iSWT | x | | | | x | | | x |
| Activity monitor given for 7 days | x | | | | | | | x (7 days prior to visit) |
| Activity monitor collected | | x | | | | | | x |
| c-PPAC score | | x | | | | | | x |
| Bronchoscopy with Chartis measurement of collateral ventilation | | x | | | | | | |
| Randomisation | | x | | | | | | |
| Treatment intervention: LVRS or BLVR | | | x | | | | | |
| Health resource utilisation (CSRI questionnaire) | | | | | x | x | x | x |

*Spirometry, gas transfer, plethysmographic lung volumes, capillary blood gases.

BLVR, bronchoscopic lung volume reduction; CAT, COPD Assessment Test; c-PPAC, clinical visit version of the PROactive Physical Activity for COPD instrument; CSRI, Client Sociodemographic and Service Receipt Inventory; FFM, fat-free mass; iSWT, incremental shuttle walk test; LVRS, lung volume reduction surgery; MRC, Medical Research Council.

Secondary outcomes will be assessed at baseline, 3 months and 12 months. Changes in respiratory-related health status (COPD Assessment Test, CAT), health-related quality of life (HRQOL) using the EQ-5D-5L, RV and fat-free mass (FFM) will be evaluated. Physical activity will be measured using the clinical visit version of the PROactive Physcal Activity in COPD instrument.

The basis for the sample size calculation is the paper by Imfeld *et al* which addressed the response to LVRS.[30] This compared the improvement in BODE score at 3 months after LVRS between survivors and non-survivors at 5 years. Long-term survivors had an improvement of 3.9 points at 3 months, the non-survivors had an improvement of 2.4 points (ie, a difference of 1.5 points). This difference is likely therefore to represent a clinically important difference between outcomes in the two LVR arms, being of sufficient magnitude to influence treatment decisions and is therefore a meaningful basis on which to power the study.

The study is powered on the basis of an average response to LVRS that will be the same as in the paper of Imfeld *et al* (ie, a fall of 3.1) and for BLVR that the improvement will be 1.6 (1.5 points less benefit). For practical purposes we have taken the BODE and the iBODE to be equivalent[29] and so, assuming an SD of 1.8 for the change in BODE score and taking a 5% significance level and 90% statistical power, 34 patients would be required in each treatment arm. Allowing for 10% dropouts we will enrol 76 individuals.

### Assignment of interventions

All participants will undergo a fibre optic bronchoscopy to allow for assessment of the presence of CV using the Chartis system.[25] This uses a balloon catheter, inflated in the target airway, with a flow sensor and a distal pressure metre. Pressure swings should continue with inspiration and expiration, but flow will diminish and then cease as the target lobe empties if there is no CV (CV −ve). If CV is present, the flow will continue (CV +ve). Subsequent participation in the trial depends on this. As the absence of CV in the target lobe is inclusion criterion, patients with CV are deemed to be screen failures and cannot proceed to randomisation (as they would not benefit from valve placement so there is no longer equipoise). However, they will still be suitable for a surgical approach and may go on to 'open label' LVRS. We will endeavour to follow these patients up at 12 months and their iBODE score will be evaluated. In some cases, CV cannot be assessed either because there is continuing low flow or an abrupt cessation in the flow trace suggesting obstruction. It may be possible to work round this by measuring the presence or absence of CV in the adjacent lobe (ie, the reciprocal position). Where CV is truly indeterminate, patients will be withdrawn from the study (ie, treated as CV positive).

If the person has a CV negative lobar target for treatment they will be immediately allocated randomly to one of the treatment arms, using an online system. Randomisation will be on a 1:1 basis based on a computer-generated random sequence with random block sizes generated by Sealed Envelope (London, UK). This will include stratification by site and by iBODE score to ensure matched numbers in both arms with an iBODE score of >7.

Patients, the study coordinator and those providing clinical care will not be blinded to treatment allocation. The assessors recording the components of the iBODE score will be blinded to reduce the potential for bias. Therefore, no unblinding process will be necessary.

### Data collection, management and analysis

At baseline a full medical history including exacerbations, hospital admissions and drug history will be taken. Pulmonary function tests (spirometry, plethysmography and gas transfer measurements) will be carried out according to the American Thoracic Society/European Respiratory Society recommendations, together with arterial blood gas analysis. Additional measures will include MRC dyspnoea score, health status (CAT), HRQOL (EQ-5D-5L) and FFM using bioimpedance. Two iSWTs will be carried out due to the learning effect seen following a test walk.[32] Figure 1 provides a summary of the trial design.

Physical activity will be assessed using the PROactive Consortium patient-reported outcome tool for physical activity in COPD.[33–35] Participants wear a DynaPort MoveMonitor accelerometer for 1 week as a measure of physical activity. At the end of this they complete the c-PPAC physical activity questionnaire which asks about the amount and difficulty of physical activity during the period covered by the monitor.

A cost-utility analysis will be carried out from a National Health Service (NHS) perspective using data from the trial. The relative value of the two approaches will be based on procedural cost, length of stay, days in hospital subsequently expressed as a cost per quality-adjusted life-year (QALY) together with data gathered using the Client Sociodemographic and Service Receipt Inventory which asks participants about their use of health and social services over the last 3 months. This will be completed at 3, 6, 9 and 12 months. They will be asked to report the frequency and intensity of their service use. Cost incurred in both arms may include: (1) medical and surgical services received as an inpatient; (2) healthcare visits in relation to COPD; and (3) COPD-related medication use. The total cost of individuals will be estimated by multiplying the resource use with national tariff used in the NHS. QALY will be estimated using EQ-5D-5L questionnaire. The five attributes of this questionnaire (mobility, self-care, usual activities, pain/discomfort/anxiety/depression) will be summarised into a single UK-derived preference-based utility score.

Outcome data analysis will be by intention to treat. Patients with missing data will be assumed to have had zero change from baseline. Unpaired t-tests will be used to compare mean response across the two groups: control and intervention. Mixed linear model procedure will be used to evaluate the effects of predefined covariables on dependent outcome variables.

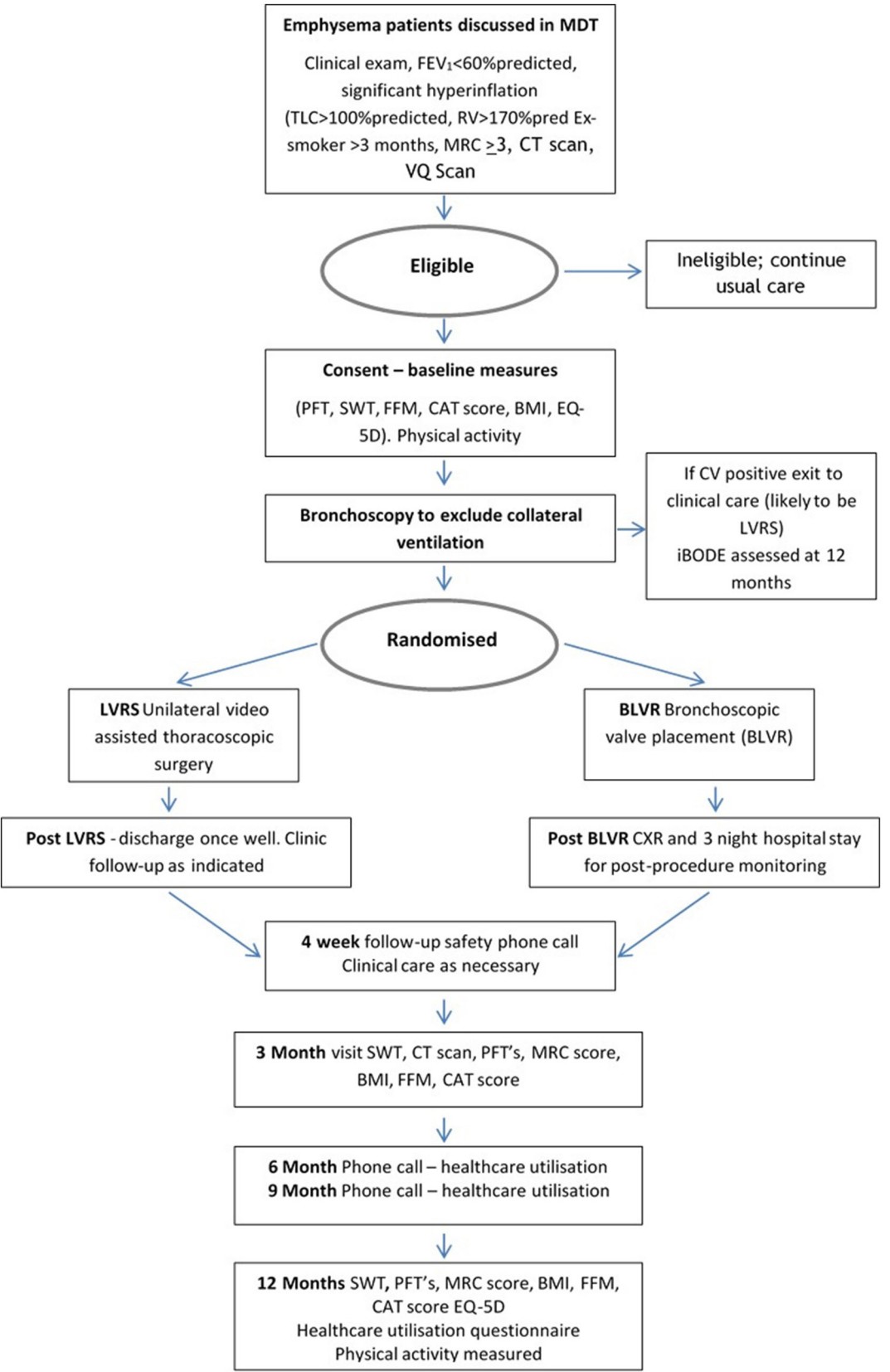

**Figure 1** Schematic outline of the trial design. BLVR, bronchoscopic lung volume reduction; BMI, body mass index; CAT, COPD Assessment Test; CV, collateral ventilation; CXR, chest X-ray; FFM, fat-free mass; iBODE, a composite score including BMI, airflow obstruction, dyspnoea and exercise capacity (incremental shuttle walk test); LVRS, lung volume reduction surgery; MDT, multidisciplinary team; MRC, Medical Research Council; PFT, pulmonary function tests, RV, residual volume; SWT, shuttle walk test; TLC, total lung capacity; VQ lung ventilation/perfusion scan.

Total costs will be calculated as the sum-product of resource use and unit cost for each patient. Incremental cost-effectiveness ratios (ICER) will be obtained by calculating the incremental costs divided by the incremental effects for the LVRS versus the BLVR group. Multivariate regression analysis will be used to adjust for baseline differences in health status between the trial groups. Uncertainty in the ICER will be addressed using bootstrapping and the estimation of cost-effectiveness acceptability curves. Estimates of ICERs will be compared with the £20 000–£30 000 per QALY threshold of cost-effectiveness recommended by the National Institute for Health and Care Excellence. No interim analyses are planned and the trial has no formal stopping guidelines.

## Patient and public involvement

Prior to commencement of the trial we held two focus groups of patients with COPD who have undergone LVRS and/or BLVR[36]. This allowed us to capture qualitative information about issues of importance to this group of patients beyond those captured by generic or disease-specific health status tools. Participants will also be invited to help with reviewing dissemination materials.

## Ethics and dissemination

Ethical approval to conduct the study has been obtained from the Fulham Research Ethics Committee, London (16/LO/0286).

Trial procedures will only begin following a clinical decision by the MDT that a patient is eligible for BLVR or LVRS. Patients will then receive verbal and written information regarding the study and informed written consent will be taken at the screening visit prior to any measurements being taken. A trial steering committee will provide overall supervision of the trial and ensures that it is being conducted in accordance with the principles of Good Clinical Practice standards.

**Acknowledgements** We thank the patient advisors who attended the focus groups and contributed to the design of this study.

**Contributors** This protocol was written by the chief investigator NSH in accordance with the sponsor's guidance for writing non-CTIMP protocols. NSH and the Research Office discussed and agreed to this study protocol. All other applicants SB, SVK, PLS, DW, SJ, JTL, WB and MCS were involved in trial design and revisions of the final protocol and will be involved in the production of the final manuscript upon trial completion. All investigators (NSH, SB, SVK, PLS, DW, SJ, MCS) agreed to perform the investigations outlined in the study protocol and to abide by this protocol except in the case of medical emergency that will be notified to the Research Office. All authors have agreed on the final manuscript for this submission.

**Funding** The study is funded by a grant to Royal Brompton and Harefield NHS Foundation Trust, from the NIHR, through the Research for Patient Benefit scheme (PB-PG-1014-35051). Imperial College, London will support the reporting of this manuscript. Trial sponsor representative: Patrik Pettersson, Royal Brompton and Harefield NHS Foundation Trust (RB&HFT), Royal Brompton Hospital.

**Competing interests** Royal Brompton has received reimbursement of clinical trial expenses from PneumRx, PulmonX, Olympus, Uptake Medical, Holaira and Creo Medical.

**Patient consent** Obtained.

**Ethics approval** Fulham NRES Committee, London (16/LO/0286).

**Provenance and peer review** Not commissioned; externally peer reviewed.

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
