## [Reviewer comments · BMJ Open]

ARTICLE DETAILS

TITLE (PROVISIONAL)	The CELEB trial: Comparative Effectiveness of Lung volume reduction surgery for Emphysema and Bronchoscopic lung volume reduction with valve placement. A protocol for a randomised controlled trial
AUTHORS	Buttery, Sara; Kemp, Samuel; Shah, Pallav; Waller, David; Jordan, Simon; Lee, John; Banya, Winston; Steiner, Michael; Hopkinson, Nicholas

VERSION 1 – REVIEW

REVIEWER	JE Hartman UMCG, The Netherlands
REVIEW RETURNED	09-Feb-2018

GENERAL COMMENTS	This is an interesting protocol to read and I am curious about the results after study completion. Three minor remarks. 1) In my opinion the following paragraphs could be left out to increase the readability: Page 10- line 24- 54 Page 11- line 36-49 (monitoring) Page 12 - line 16-41 2) according to the clinical trial register, the trial already started in 2016, authors should mention this in the manuscript 3)Page 5- line 9: check whether reference 10 is correct (and other references could be used).
---

REVIEWER	Michael Perch Righospitalet Heartcentre Blegdamsvej 9 2100 Copenhagen Denmark Department of Clinical Medicine, University of Copenhagen, Copenhagen, Denmark
REVIEW RETURNED	23-Feb-2018

GENERAL COMMENTS	This study is trying to answer an important question within advanced emphysema therapy, namely if there is any treatment efficacy between Bronchoscopic Lung Volume Reduction (BLVR) and Lung Volume Reduction Surgery (LVRS). LVRS perceived as the gold standard in volume reduction therapy. .Overall the study is well made and the manuscript describing the study protocol is suitable for publication, although there are some minor details which need clarification. Methods and analysis The patient screening process needs to be further described before inclusion in the study. The MDT conference and the way the patients arrive at MDT. The study group have chosen inclusion criteria which are somewhat
---

	different from the latest values from “Endoscopic Lung Volume Reduction: An Expert Panel Recommendation-Update 2017” recommending FEV1<50%, RV >175% and also RV/TLC ratio above 0.58. see attached file. Please comment on this How does the study group plan to define heterogeneous emphysema? The LVRS procedure will be done by discretion of the surgeon. This approach does reflect the clinical reality, but the study group might consider classifying the possible LVRS procedures (ex. Horseshoe, lobectomy, multiple smaller wedges ect.) or estimate the total removed lung volume? Page 6 linie 29: What is HDU? Page 7: linie 24: Changes in health related quality of life will be measured with CAT score. Is CAT score valid for HRQL? On page 9 EQ-5D is mentioned ? Page 10 Linie 7 EQ-5D or CAT? Why FFM? Is 10% for dropouts sufficient? Page 8, Linie 16-37. Pitfalls using the Chartis measurement is well described, even so measurements can be inconclusive. Will these patients be excluded? Page 8, linie 24 If these patients do... this sentence is unclear and needs to be clarified. Page 11, linie 37 Monitoring. GCP? Page 12, Linie 12. Do the patients give informed consent before the MDT? (see first comment about Methods and analysis) Clarification please... Page 20. Include MDT in the table where is the informed consent obtained? See previous comments
--	--

REVIEWER	Prof. Dr. Ralf Eberhardt MD Thoraxklinik at the University of Heidelberg, Germany
REVIEW RETURNED	01-Mar-2018

GENERAL COMMENTS	Dear Editor, thank you for inviting me to review the manuscript “The CELEB trial: Comparative Effectiveness of Lung volume reduction surgery for Emphysema and Bronchoscopic lung volume reduction with valve placement: a protocol for a randomised controlled trial” from S. Buttery et al. This is a study protocoll describing a randomized controlled trial comparing endoscopic valve placement vs. lung volume reduction surgery in patients with severe emphysema. The manuscript is well written and contains a clear explanation of the study design. Therefore I have no comments about the manuscript. However, it would be useful to discuss whether the comparison of endoscopic and surgical treatment of severe emphysema is advisable, because these are complementary therapies for different patients. But this is not the question to the reviewer of this manuscript. Therefore I would recommend adding an editorial to the manuscript.
---

REVIEWER	Babu Naidu University of Birmingham
REVIEW RETURNED	11-Mar-2018

GENERAL COMMENTS	The research question is worthwhile but i am afraid the proposed study will fail to answer the question 1. The gold standard for lung volume reduction surgery is bilateral surgery . Almost all the references relate to bilateral surgery ,
---

	Calculation for sample size are based on bilateral LVRS studies .The proposed study chooses to compare only unilateral LVRS. Theoretically there are advantages to bilateral surgery which will be lost in this study and the risk is that the findings will be extrapolated to bilateral LVRS 2. The proposed outcome index i-BODE differs from the calculations used for sample size which is based on the BODE index . This differs in the form of the type of exercise test. Shuttle walk test versus six minute walk test . This is not discussed .no refernces comparing the 2 3. The study is underpowered The basis for calculation of difference in outcome in BODE index is made on a 3 month observation in the imfield et al paper (3.2 difference) but the investigators outcome measure is at 12 months . Ginsburg ME, at al . (LVRS using the NETT selection criteria. Ann Thorac Surg. 2011 May;91(5):1556-60;) showed a more modest change of -2.3 (±1.5) in bode after bilateral LVRS at 12 months Citing the Transform data for EBV 1.8 at 6 months (CI 0.9 -2.6)(Vol. 196, No. 12 Dec 15, 2017) would have been appropriate. This the proposed difference is unrealistic 2. the explanation of 1.5 difference being clinical significant is flawed. The study cited showed that post LVRS treatment difference of 1.5 at 3 months is associated with ong term survival . Extrapolating these findings to 12 months is problematic and assuming that pre treatment values have the same prediction is mistaken 3. a better citation is Utku, et al European Respiratory Journal 2013 in terms of clinically significant difference in iBODE
--	--

VERSION 1 – AUTHOR RESPONSE

Reviewer: 1

Comments to the Author:

1) In my opinion the following paragraphs could be left out to increase the readability:

Page 10- line 24- 54

Page 11- line 36-49 (monitoring)

Page 12 - line 16-41

Paragraphs removed from manuscript.

2) According to the clinical trial register, the trial already started in 2016, authors should mention this in the manuscript

Comment addressed in manuscript; page 4.

The CELEB study is a multi-centre, randomised - controlled parallel group trial that opened for recruitment in October 2016.

3) Page 5- line 9: check whether reference 10 is correct (and other references could be used). Reference 10 removed and new reference added.

Kemp SV, Slebos DJ, Kirk A, et al. A Multicenter Randomized Controlled Trial of Zephyr Endobronchial Valve Treatment in Heterogeneous Emphysema (TRANSFORM). American journal of respiratory and critical care medicine 2017;196(12):1535-43. doi: 10.1164/rccm.201707-1327OC [published Online First: 2017/09/09]

Reviewer: 2

Comments to the Author:

1. The patient screening process needs to be further described before inclusion in the study. The MDT conference and the way the patients arrive at MDT.

Comment addressed; pages 5

Patients fitting these criteria will be identified from outpatient clinics and put forward for discussion at the MDT meeting.

The MDT will make a decision on whether a patient is suitable for both interventions and if so, that there is equipoise between the treatments options. It is only after this is agreed that the screening visit will be arranged and consent taken.

2. The study group have chosen inclusion criteria which are somewhat different from the latest values from "Endoscopic Lung Volume Reduction: An Expert Panel Recommendation-Update 2017" recommending FEV1<50%, RV >175% and also RV/TLC ratio above 0.58. see attached file. Please comment on this

This protocol was devised by a panel of experts in the field of LVR and received approval in 2016, prior to the above manuscript being published.

3. How does the study group plan to define heterogeneous emphysema?

We have not defined heterogeneous emphysema in this protocol as the decision is based on review by a multidisciplinary team. The initial assessment is based on visual inspection of the CT together with emphysema scoring and evaluation of fissure integrity. Participants may also undergo perfusion scan.

4. The LVRS procedure will be done by discretion of the surgeon. This approach does reflect the clinical reality, but the study group might consider classifying the possible LVRS procedures (ex. Horseshoe, lobectomy, multiple smaller wedges ect.) or estimate the total removed lung volume?

The type of operation will be presented as part of the descriptive data.

5. Page 6 linie 29: What is HDU?

HDU refers to High Dependency Unit. This has now been clarified in the text on page 5.

6. Page 7: linie 24: Changes in health related quality of life will be measured with CAT score. Is CAT score valid for HRQL? On page 9 EQ-5D is mentioned ? Page 10 Linie 7 EQ-5D or CAT?

Thank you for highlighting this. The EQ-5D-5L will be used to measure HRQOL and the CAT, as a general measure of the impact of COPD on the patient's health status.

The text has been amended to clarify this on pages 6 and 8.

Changes in respiratory related health status (COPD assessment test (CAT)), health related quality of life (HRQOL) using the EQ-5D-5L.

health status (CAT), HRQOL (EQ-5D-5L)

7. Why FFM?

There is evidence that skeletal muscle wasting is a feature of COPD, in particular emphysema, which may respond to lung volume reduction interventions.

8. Is 10% for dropouts sufficient?

This was estimated based on our clinical/trial experience

9. Page 8, Line 16-37. Pitfalls using the Chartis measurement is well described, even so measurements can be inconclusive. Will these patients be excluded?

This has already been addressed in the manuscript.

"Where CV is truly indeterminate, patients will be withdrawn from the study (i.e. treated as CV positive)".

10. Page 8, line 24 if these patients do... this sentence is unclear and needs to be clarified. Comment addressed; page 7.

However they will still be suitable for a surgical approach and may go on to "open label" LVRS. We will endeavour to follow these patients up at 12 months and their iBODE score will be evaluated

11. Page 11, line 37 Monitoring. GCP?

Comment addressed; page 12

A trial steering committee will provide overall supervision of the trial and ensures that it is being conducted in accordance with the principles of Good Clinical Practice (GCP) standards.

12. Page 12, Line 12. Do the patients give informed consent before the MDT? (see first comment about Methods and analysis) Clarification please...

Comment addressed; page 12

Patients will then receive verbal and written information regarding the study and informed written consent will be taken at the screening visit prior to any measurements being taken.

13. Page 20. Include MDT in the table where is the informed consent obtained?

See previous comments

Reviewer 3:

This is a study protocol describing a randomized controlled trial comparing endoscopic valve placement vs. lung volume reduction surgery in patients with severe emphysema. The manuscript is well written and contains a clear explanation of the study design. Therefore I have no comments about the manuscript.

However, it would be useful to discuss whether the comparison of endoscopic and surgical treatment of severe emphysema is advisable, because these are complementary therapies for different patients. But this is not the question to the reviewer of this manuscript. Therefore I would recommend adding an editorial to the manuscript.

Thank you. We do think that the procedures should be compared in the population in the present study where there is equipoise

Reviewer 4:

1. The gold standard for lung volume reduction surgery is bilateral surgery. Almost all the references relate to bilateral surgery, Calculation for sample size are based on bilateral LVRS studies. The proposed study chooses to compare only unilateral LVRS. Theoretically there are advantages to bilateral surgery which will be lost in this study and the risk is that the findings will be extrapolated to bilateral LVRS

We do not agree that bilateral surgery can be said to be the gold standard.

2. The proposed outcome index i-BODE differs from the calculations used for sample size which is based on the BODE index . This differs in the form of the type of exercise test. Shuttle walk test versus six minute walk test. This is not discussed .no refernces comparing the 2 The study is underpowered The basis for calculation of difference in outcome in BODE index is made on a 3 month observation in the imfield et al paper (3.2 difference) but the investigators outcome measure is at 12 months . Ginsburg ME, et al . (LVRS using the NETT selection criteria. Ann Thorac Surg. 2011 May;91(5):1556-60;) showed a more modest change of -2.3 (\pm 1.5) in bode after bilateral LVRS at 12 months Citing the Transform data for EBV 1.8 at 6 months (CI 0.9 -2.6)(Vol. 196, No. 12 | Dec 15, 2017) would have been appropriate. This the proposed difference is unrealistic. The explanation of 1.5 difference being clinical significant is flawed. The study cited showed that post LVRS treatment difference of 1.5 at 3 months is associated with long term survival. Extrapolating these findings to 12 months is problematic and assuming that pre-treatment values have the same prediction is mistaken. A better citation is Utku, et al European Respiratory Journal 2013 in terms of clinically significant difference in iBODE.

http://erj.ersjournals.com/content/42/Suppl_57/P2230

Thank you. The BODE and i-BODE have very similar characteristics. The alternative citation suggested is from a conference abstract which found a 1.2 point fall in i-BODE after pulmonary rehabilitation. Although a difference between treatment arms based on the same effect size as a PR programme (1.2 points) would have been a reasonable alternative to the one we have selected, we are comfortable with the choice of (1.5 points), which is a difference in the score associated with improved survival, as a clinically significant outcome.

Comments to the Author

We have amended the title as advised.

The CELEB trial: Comparative Effectiveness of Lung volume reduction surgery for Emphysema and Bronchoscopic lung volume reduction with valve placement. A protocol for a randomised controlled trial

References

Kemp SV, Slebos DJ, Kirk A, et al. A Multicenter Randomized Controlled Trial of Zephyr Endobronchial Valve Treatment in Heterogeneous Emphysema (TRANSFORM). American journal of respiratory and critical care medicine 2017;196(12):1535-43. doi: 10.1164/rccm.201707-1327OC [published Online First: 2017/09/09]

VERSION 2 – REVIEW

REVIEWER	Jorine Hartman UMCG hospital, The Netherlands
REVIEW RETURNED	07-May-2018

GENERAL COMMENTS	No further comments.
----------------------

REVIEWER	Michael Perch Rigshospitalet, Denmark
REVIEW RETURNED	26-May-2018

GENERAL COMMENTS	The manuscript did benefit from the revision and the protocol is now much easier to read. I recommend the manuscript for publication, although I have a minor comment: You choose to measure treatment efficacy using changes in iBODE extrapolating from predictive results from BODE? But that has not been shown? (Page 4, line 42) Similar with the calculation of study power which is based on changes in BODE (SD 1,8), not iBODE. Although similar index's they are not exactly the same. please comment Page 4 linie 50 CONSORT?
--

VERSION 2 – AUTHOR RESPONSE

Dear Assistant Editor and reviewers,

Thank you for the suggested minor revisions and recommendation for publication.

Attached is a revised manuscript with ammendments along with a clean copy of the revised manuscript.

We hope that the manuscrpit is now acceptable for publication.

- The author provided a marked copy. Please contact the publisher for full details.